# Aldh inhibitor restores auditory function in a mouse model of human deafness

**Guang-Jie Zhu**[1☯]**, Sihao Gong**[1☯]**, Deng-Bin Ma**[1☯]**, Tao Tao**[1☯]**, Wei-Qi He**[1,2☯]**,
Linqing Zhang**[1]**, Fang Wang**[1]**, Xiao-Yun Qian**[1]**, Han Zhou**[1]**, Chi Fan**[1]**, Pei Wang**[1]**,
Xin Chen**[1]**, Wei Zhao**[1]**, Jie Sun**[1]**, Huaqun Chen**[3]**, Ye Wang**[4]**, Xiang Gao**[1]**, Jian Zuo**[5]**,
Min-Sheng Zhu**[1,6]***, Xia Gao**[1]***, Guoqiang Wan**[1,6]***

**1** Department of Otorhinolaryngology, Provincial Key Discipline of the affiliated Drum Tower Hospital of
Nanjing University and Model Animal Research Center, MOE Key Laboratory of Model Animal for Disease
Studies, School of Medicine, Nanjing University, Nanjing, China, **2** Jiangsu Key Laboratory of
Neuropsychiatric Diseases and Cambridge-Suda (CAM-SU) Genomic Resource Center, Medical College of
Soochow University, Suzhou, China, **3** College of Life Science, Nanjing Normal University, Nanjing, China,
**4** Nanjing MuCyte Biotechnology Co., Ltd., Nanjing, China, **5** Department of Biomedical Sciences, School of
Medicine, Creighton University, United States of America, **6** Institute for Brain Sciences, Nanjing University,
Nanjing, China

☯ These authors contributed equally to this work.
* zhums@nju.edu.cn (MSZ); xiagaogao@hotmail.com (XG); guoqiangwan@nju.edu.cn (GW)

doi.org/10.1371/journal.pgen.1009040

Health, UNITED STATES

**Data Availability Statement:** The RNA-seq data
are available at GEO database with accession
number GSE139145.

**Funding:** The research is supported by the National
Natural Science Funding of China (Project IDs:

## Abstract

Genetic hearing loss is a common health problem with no effective therapy currently avail-
able. DFNA15, caused by mutations of the transcription factor POU4F3, is one of the most
common forms of autosomal dominant non-syndromic deafness. In this study, we estab-
lished a novel mouse model of the human DFNA15 deafness, with a *Pou4f3* gene mutation
(*Pou4f3Δ*) identical to that found in a familial case of DFNA15. The *Pou4f3(Δ/+)* mice suf-
fered progressive deafness in a similar manner to the DFNA15 patients. Hair cells in the
*Pou4f3(Δ/+)* cochlea displayed significant stereociliary and mitochondrial pathologies, with
apparent loss of outer hair cells. Progression of hearing and outer hair cell loss of the
*Pou4f3(Δ/+)* mice was significantly modified by other genetic and environmental factors.
Using *Pou4f3(-/+)* heterozygous knockout mice, we also showed that DFNA15 is likely
caused by haploinsufficiency of the *Pou4f3* gene. Importantly, inhibition of retinoic acid sig-
naling by the aldehyde dehydrogenase (Aldh) and retinoic acid receptor inhibitors promoted
Pou4f3 expression in the cochlear tissue and suppressed the progression of hearing loss in
the mutant mice. These data demonstrate *Pou4f3* haploinsufficiency as the main underlying
cause of human DFNA15 deafness and highlight the therapeutic potential of Aldh inhibitors
for treatment of progressive hearing loss.

## Author summary

More than 50% of deafness cases are due to genetic defects with no treatment available.
DFNA15, caused by mutations of the transcription factor *POU4F3*, is one of the most
common types of autosomal dominant non-syndromic deafness. Here, we established a
novel mouse model with the exact *Pou4f3* mutation identified in human patients. The

81371090, 31330034, 31671548 to MSZ and Project ID: 31771153 to GW; www.nsfc.gov.cn). The study is also funded by the National Institutes of Health (Project ID: R01DC006471 to JZ; www. nih.gov). The funders had no role in study design, data collection and analysis, decision to publish, or preparation of the manuscript.

**Competing interests:** The authors have declared that no competing interests exist.

mutant mouse display similar auditory pathophysiology as human patients and exhibit multiple hair cell abnormalities. The onset and severity of hearing loss in the mouse model is highly modifiable to environmental factors, such as aging, noise exposure or genetic backgrounds. Using a new knockout mouse model, we found Pou4f3 haploinsufficiency as the underlying mechanism of human DFNA15. Importantly, we identified Aldh inhibitor as a potent small molecule for upregulation of Pou4f3 and treatment of hearing loss in the mutant mouse. The identification of Aldh inhibitor for treatment of DFNA15 deafness represents a major advance in the unmet medical need for this common form of progressive hearing loss.

## Introduction

Hearing loss is one of the most common sensory defects resulting from both genetic and environmental insults that affect more than 250 million people worldwide [1]. Despite the prevalence of hearing loss, no Food and Drug Administration-approved therapeutic treatment is currently available. Among the genetic hearing disorders, 20% of the cases are the autosomal dominant inheritance of nonsyndromic hearing loss [2]. *POU4F3* mutations are associated with DFNA15, one of the common autosomal dominant forms of progressive hearing loss [3–5]. Studies using the *Pou4f3-null* mice indicate that *Pou4f3* plays an essential role in differentiation and survival of inner ear hair cells [6–8]. However, how Pou4f3 mutations in human patients cause the dominant form of progressive hearing loss remains unknown.

To shed light into the pathology and molecular basis of DFNA15 progressive hearing loss, we generated a novel mouse model harboring the exact human *POU4F3* mutation identified in the first identified familial case of DFNA15 [3]. In this study, using a combination of Pou4f3 mutant and knockout animal models, we examined the pathophysiology and hair cell histopathology of DFNA15 deafness, revealed *Pou4f3* haploinsufficiency as the main underlying cause of the deafness, and identified retinoic acid signaling inhibitors for *Pou4f3* upregulation and treatment of DFNA15 deafness.

## Results

### The DFNA15 mouse model displays progressive hearing loss

We established a knock-in mouse line with an 8bp deletion and a C-T reversion of the mouse *Pou4f3* gene (*Pou4f3Δ*; S1A and S1B Fig), which was identical to the *POU4F3* mutation found in an Israeli Jewish DFNA15 patient family [3]. This mutation of *Pou4f3* in the target ES cells was confirmed by Southern blot analysis and sequencing (S1C Fig). Genotyping analysis of *Pou4f3(Δ/+)* mice with tail genomic DNA showed a specific PCR product (542 bp), suggesting successful germ line transmission of the mutation (S1D Fig). Previous *in vitro* studies demonstrate that the 8bp deletion of *Pou4f3* gene results in a truncated Pou4f3 protein with impaired nuclear localization [9, 10]. Western blot of P3 cochleae from wildtype (+/+), heterozygous (Δ/+) and homozygous (Δ/Δ) cochlea also showed a smaller band of the truncated Pou4f3 protein in the cochlea of *Pou4f3(Δ/+)* and *Pou4f3(Δ/Δ)* mice (S1E Fig). When we expressed wildtype Pou4f3 (Pou4f3-wt) and the mutant Pou4f3 (Pou4f3-8del) in HeLa cells, mutant Pou4f3 also displayed impaired nuclear localization (S1F Fig). Similarly, Pou4f3 immunostaining of P10 wildtype and mutant cochleae showed cytoplasmic localization of the mutant Pou4f3 and weaker nuclear Pou4f3 signal in outer hair cells of the *Pou4f3(Δ/+)* cochleae (S1G Fig).

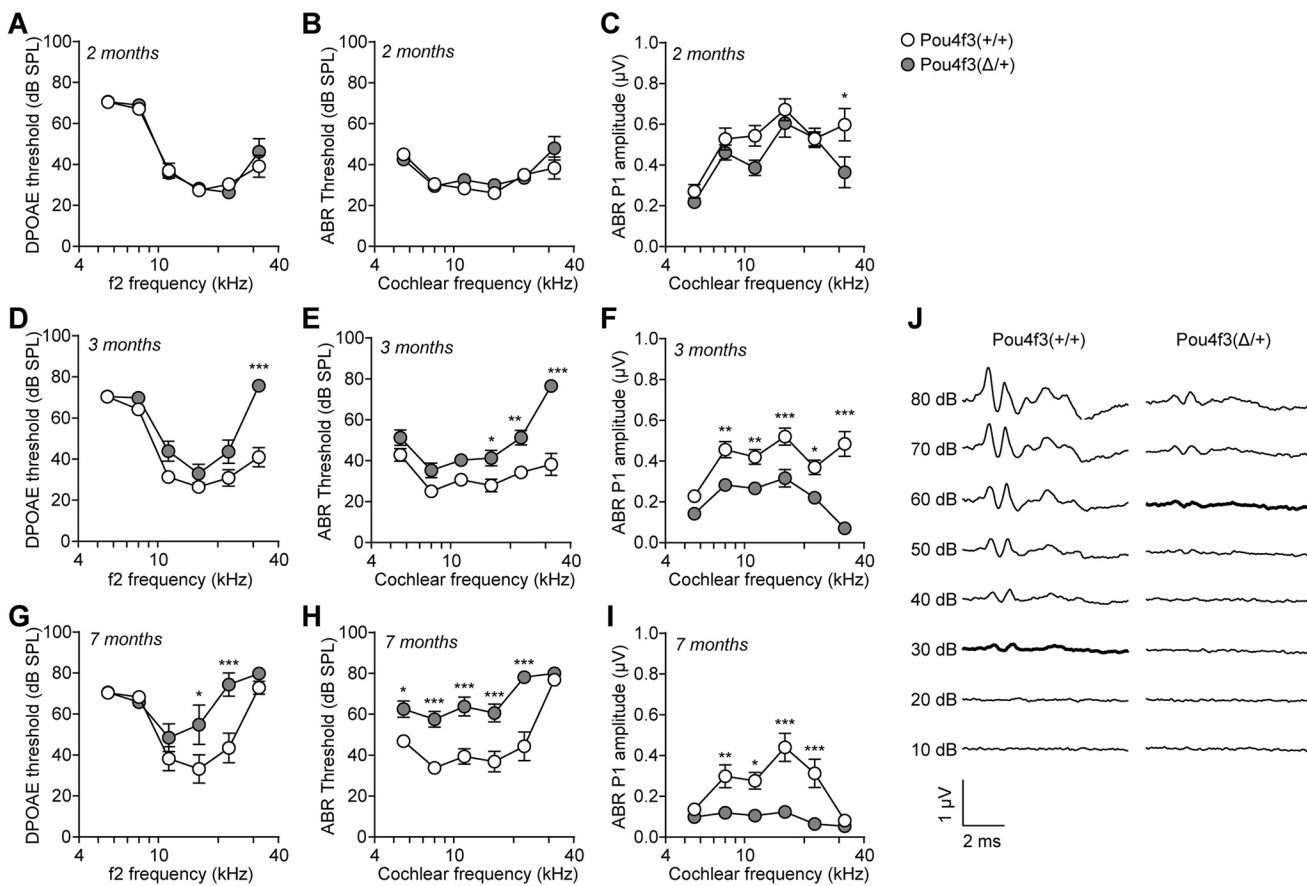

**Fig 1. *Pou4f3(Δ/+)* mice display progressive hearing loss.** (**A**-**C**) 2 months (n = 9–10), (**D**-**F**) 3 months (n = 14–16) and (**G**-**I**) 7 months (n = 8 each) old wildtype *Pou4f3(+/+)* and mutant *Pou4f3(Δ/+)* mice were tested with DPOAE and ABR. (**A, D, G**) DPOAE thresholds; (**B, E, H**) ABR thresholds; (**C, F, I**) ABR peak 1 (P1) amplitudes. * P < 0.05, ** P < 0.01 and *** P < 0.001 by two-way ANOVA. (**J**) ABR waveforms of 3 months old *Pou4f3(+/+)* and *Pou4f3 (Δ/+)* mice at 32 kHz. Data were mean of n = 14–16 animals. ABR thresholds of each genotype were highlighted. dB, decibel; SPL, sound pressure level.

Human DFNA15 deafness presents as post-lingual progressive hearing loss [3, 11]. We then measured auditory functions of 2–7 months old *Pou4f3(Δ/+)* mice using distortion product otoacoustic emissions (DPOAEs) and auditory brainstem responses (ABRs). Both DPOAE (Fig 1A) and ABR thresholds (Fig 1B) of *Pou4f3(Δ/+)* mice were comparable to the wildtype mice at 2 months age. However, ABR peak 1 (P1) amplitudes of the *Pou4f3(Δ/+)* mice were significantly reduced at 32 kHz (Fig 1C and S2 Fig), indicating that these mutant mice start to display mild auditory phenotype at 2 months. As mice aged further to 3 months (Fig 1D–1F and 1J) and 7 months (Fig 1G–1I), DPOAE thresholds, ABR thresholds and ABR P1 amplitudes of the *Pou4f3(Δ/+)* mice altered significantly and progressively from high to low cochlear frequencies (Fig 1 and S2 Fig). This observation suggests that *Pou4f3(Δ/+)* mice displayed an adult-onset progressive hearing loss that starts at high frequencies, consistent with the progression of human DFNA15 [11]. In contrast to the wildtype or heterozygous *Pou4f3(Δ/+)* mice, the homozygous *Pou4f3(Δ/Δ)* were profoundly deaf (S3A Fig) and lost all hair cells (S3B Fig) within the first month of age. This remarkable phenotype resembles that observed in *Pou4f3* knockout mice [12, 13], suggesting that hair cell survival is critically dependent on *Pou4f3* function. Our result also indicates that *Pou4f3* plays a dosage-dependent role in auditory function.

As some DFNA15 patients also complained about vestibular problems (S1 Table), we examined the balance functions of the *Pou4f3(Δ/+)* mice using rotarod tests. Both 7 months old wildtype and *Pou4f3(Δ/+)* mice were subjected to 3 test protocols: 12 rpm, 20 rpm constant speed and 0–44 rpm accelerating speed (S4A Fig). Results from all the 3 test protocols showed that *Pou4f3(Δ/+)* mice had similar vestibular function as the wildtype littermates (S4B Fig). The hearing loss phenotype of the *Pou4f3(Δ/+)* mice was confirmed by DPOAE tests (S4C Fig). To examine the effects of Pou4f3 mutation on vestibular hair cells, we then dissected utricular sensory epithelia from the wildtype and *Pou4f3(Δ/+)* mice for immunostaining of hair cell markers (S4D Fig). We found that the densities of hair cells in both extrastriolar and striolar regions were comparable between wildtype and *Pou4f3(Δ/+)* mice (S4E and S4F Fig). Interestingly, the sizes of utricular sensory epithelia from *Pou4f3(Δ/+)* mice appeared to be smaller than those from the wildtype mice (S4G Fig), indicating mild but significant effect of Pou4f3 mutation on vestibular sensory epithelia. The mild vestibular phenotype of the *Pou4f3 (Δ/+)* mice was consistent with the clinical data, where vestibular complains were not common among the DFNA15 patients and vestibular tests appeared normal even in patients with such complains (S1 Table). We have previously shown that age-related vestibular dysfunction occurs much later than hearing loss [14]. It is possible that the *Pou4f3(Δ/+)* mice may develop vestibular deficits at a much older age, and the compensatory effects of central vestibular system and other sensory modalities [15] may explain the mild and inconsistent vestibular impairments observed in both DFNA15 patients and mouse models.

## The DFNA15 mouse model displays multiple hair cell pathologies

To address the pathological basis of the DFNA15 deafness, we performed thorough histological analyses in *Pou4f3(Δ/+)* mice at 4–6 months age (Fig 2 and S5 Fig). The cross-sectioned *Pou4f3(Δ/+)* cochleae stained with hematoxylin/eosin (H&E) showed clear overall structures including tectorial membranes, inner hair cells (IHCs), and supporting cells; however, some outer hair cells (OHCs) were missing at 32 kHz cochlear region (Fig 2A). Selective degeneration of OHCs but not IHCs at 32 kHz was then confirmed by immunostaining of the sensory epithelia with the hair cell marker Myo7a (Fig 2B and 2C and S5 Fig).

To further examine if the ultrastructures of IHCs and the surviving OHCs were morphologically normal, we performed SEM analyses of 6 months old wildtype and *Pou4f3(Δ/+)* cochleae. SEM of the wildtype organ of Corti showed a clear row of IHC and three rows of OHCs, while many OHCs were lost in middle-basal turn of *Pou4f3(Δ/+)* cochlea and replaced with filled flat epithelium (Fig 2D). This data is consistent with OHC degeneration observed in Myo7a-stained *Pou4f3(Δ/+)* cochlear whole mounts (Fig 2B and 2C). Interestingly, the remainder OHCs appeared to have less stereocilia bundles in *Pou4f3(Δ/+)* cochleae compared to wildtype (Fig 2E). Quantitative results showed that the number of OHC stereocilia from *Pou4f3(Δ/+)* mice was significantly less than that of wildtype at both cochlear turns (Fig 2F). Additionally, although the IHCs of the 6 months old *Pou4f3(Δ/+)* cochlea did not show significant degeneration, their stereocilia displayed remarkable pathology featured by fusion and elongation of the stereociliary bundles (Fig 2D). Based on the SEM images, we then measured the lengths of the stereocilia above the reticular lamina and found significant elongations of IHC stereociliary bundles at both turns of *Pou4f3(Δ/+)* cochleae (Fig 2G).

To probe possible degenerative events inside the hair cells, we also examined both OHCs and IHCs by TEM analyses. Hair cells of the *Pou4f3(Δ/+)* mice had normal nuclei, cuticular plate and overall cellular morphologies (Fig 2H and 2J). However, both OHCs and IHCs of the *Pou4f3(Δ/+)* cochlea showed significant reductions in mitochondrial density (Fig 2H-K) and displayed mitochondrial vacuolization (Fig 2H and 2J).

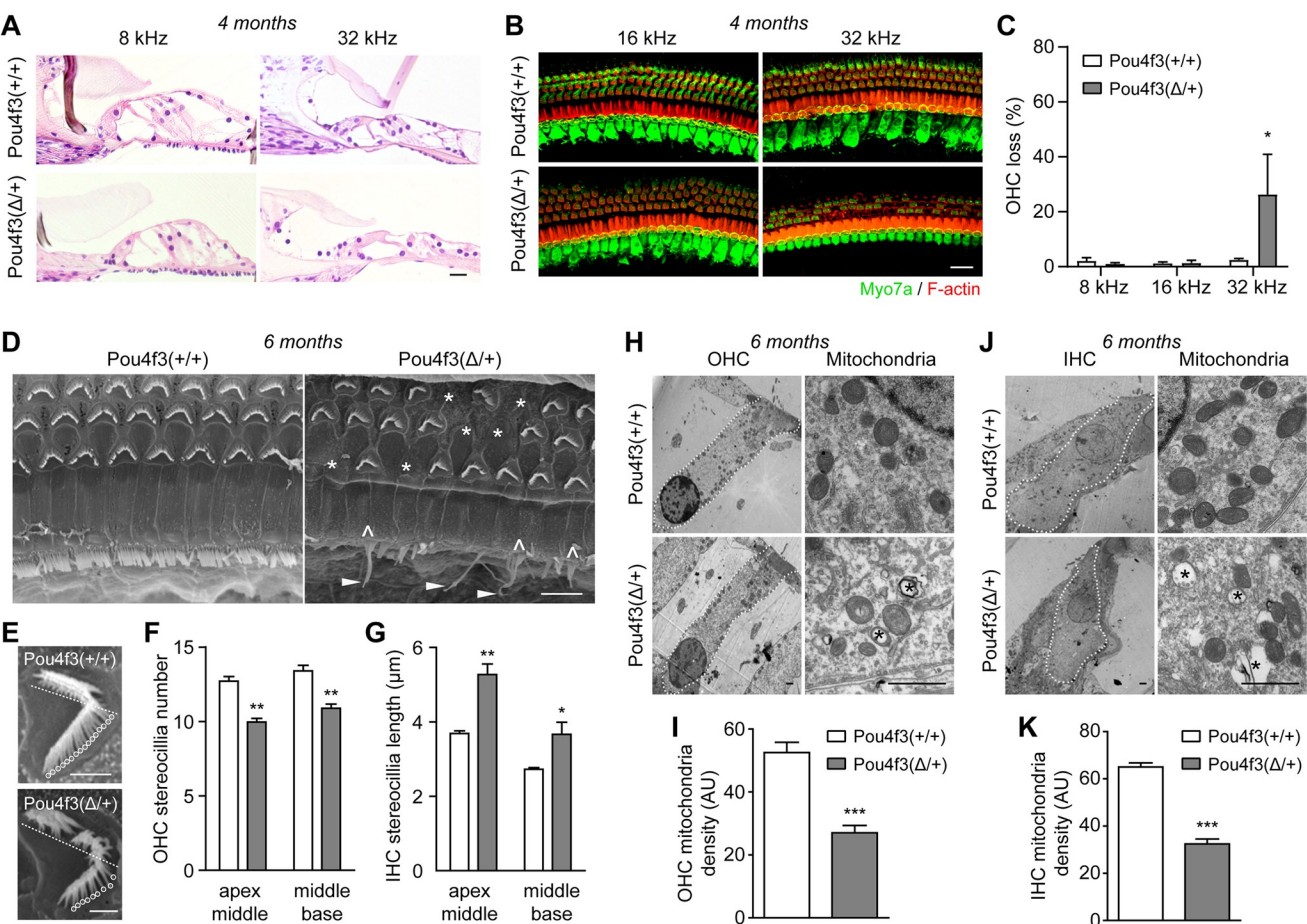

**Fig 2. *Pou4f3(Δ/+)* mice display degeneration of outer hair cells and defects in stereocilia and mitochondria of the surviving hair cells.** (**A**) H&E staining images of organ of Corti plastic sections from 4 months old wildtype and *Pou4f3(Δ/+)* mice. (**B**) Myo7a immunostaining images of the cochlear sensory epithelia from 4 months old wildtype and *Pou4f3(Δ/+)* mice. Hair cells and F-actin was labelled with Myo7a (green) and Rhodamine-phalloidin (red), respectively. (**C**) Percentage of outer hair cell loss in wildtype and *Pou4f3(Δ/+)* mice. * P < 0.05 by two-way ANOVA, n = 4–5 cochleae of each genotype. (**D**) SEM images of middle-base turn cochlear sensory epithelia from 6 months old wildtype and *Pou4f3(Δ/+)* mice. Missing OHCs (*), fused stereocilia (^) and elongated stereocilia (arrow heads) of IHCs were indicated in *Pou4f3(Δ/+)* cochlea. (**E**) Evaluation of OHC stereocilia number in wildtype (top panel) and *Pou4f3(Δ/+)* cochleae (bottom panel). Half of the "V" shaped stereocilia bundles (bellow the dashed lines) and roots of the stereocilia were counted for each OHC (circles). (**F**) Quantification of OHC stereocilia density in both wildtype and *Pou4f3(Δ/+)* cochleae at two cochlear half turns (apex-middle and middle-base). ** P < 0.01 by unpaired student's t-test, n = 10–23 images from 4 cochleae of each genotype. (**G**) Quantification of IHC stereocilia lengths in wildtype and *Pou4f3(Δ/+)* cochleae at the two cochlear half turns. * P < 0.05 and ** P < 0.01 by unpaired student's t-test, n = 4–7 images from 4 cochleae of each genotype. (**H, J**) TEM images of OHCs (**H**) and IHCs (**J**) from 6 months old wildtype and *Pou4f3(Δ/+)* cochleae (middle turn). OHC or IHC was delineated with dashed lines. High magnification images showing mitochondria from OHC (**H**) or IHC (**J**) were also displayed. Asterisks (*) indicate vacuolization of the mitochondria. (**I, K**) Quantification of mitochondria density in (**I**) OHCs and (**K**) IHCs from wildtype and *Pou4f3(Δ/+)* cochleae. *** P < 0.01 by unpaired student's t-test, n = 4–5 images from three mice of each genotype. Scale bars were 20 μm (**A-B**), 5 μm (**D**) and 1 μm (**E, H, J**).

The diverse histopathologies in hair cell stereocilia, mitochondria and OHC survival of the *Pou4f3(Δ/+)* mice indicate that the *Pou4f3* mutation may affect its transcriptional regulation of genes involved in multiple pathways. We first addressed whether *Pou4f3* mutation may affect the expression of known downstream targets, including *Lhx3* [8], *Gfi1* [7], *Bdnf* [6], *Ntf3* [6], *Myo6* [16], *Caprin1* [17] and *Nr2f2* [18]. We collected cochlear samples from both wildtype and *Pou4f3(Δ/+)* mutant mice at age of 2 months, when the mutant mice displayed mild hearing loss without significant hair cell loss that may confound gene expression analyses. To our surprise, while the wildtype *Pou4f3* was downregulated in the *Pou4f3(Δ/+)* cochlea, none of its

known targets showed change in expression by RT-qPCR analyses (S6A Fig). Bulk RNA-seq analyses of the wildtype and *Pou4f3(Δ/+)* cochleae showed altered gene expressions in biologically processes including metabolism, biosynthesis, and sensory perceptions (S2 Table and S6B Fig). Some of the gene expression changes were validated by RT-qPCR, including *Tmem237* and *Sra1* (S6C Fig). Interestingly, 13 of the top 20 gene ontology pathways were related to cellular metabolic processes (S6B Fig). This result is consistent with mitochondrial abnormalities observed in *Pou4f3(Δ/+)* hair cells (Fig 2H–2K) and implies that Pou4f3 may play a homeostatic role in hair cell mitochondria and cellular metabolism.

## The onset of DFNA15 deafness is modified by genetic background and noise exposure

The audiological characteristics were highly variable among individual DFNA15 patients carrying the Pou4f3-8del mutation [11]. We hypothesize that other genetic or environmental factors may modify the onset or progression of DFNA15 deafness. The C57BL/6J mouse displays progressive hearing loss caused by a strain-specific homozygous *Cdh23(ahl)* mutation [19]. To reduce the influence of *Cdh23(ahl)* mutation, we crossed the mutant *Pou4f3(Δ/+)* mice (C57BL/6J background) with FVBN mice to produce both wildtype and mutant mice on a mixed genetic background. We then measured auditory functions in 3–12 months old *Pou4f3 (Δ/+)* mice using DPOAEs and ABRs. At 3 weeks age, both ABR and DPOAE thresholds were comparable between wildtypes and heterozygous mutants; however, the homozygous mutants were completely deaf (S7A and S7B Fig), similar to the 1 months old homozygous mutants on C57BL/6J background. These results indicate that the essential roles of Pou4f3 in hair cell survival and function were independent of genetic backgrounds. In addition, DPOAE thresholds, ABR thresholds and ABR P1 amplitudes were comparable between wildtype and mutant mice even at 3 months (S7C–S7E Fig) and 6 months (S7F–S7H Fig), except for slight elevations of DPOAE thresholds at 8 kHz in the mutant mice. However, by 12 months age, elevations in DPOAE (S7I Fig) and ABR thresholds (S7J Fig) at high frequencies and reductions in ABR P1 amplitudes at most frequencies (S7K Fig) became apparent in the mutant mice. Consistently, significant loss of OHCs at 32 kHz was observed in 12 months old mutant mice (S7L and S7M Fig). This data was in contrast with the severe hearing loss phenotype in the mutant mice on C57BL/6J background observed as early as 3 months age (Fig 1), suggesting that the onset of hearing loss in the *Pou4f3(Δ/+)* mice was dependent on the genetic background or predispositions.

We then subjected the 4 months old wildtype and mutant mice on a mixed background, who had not developed hearing loss, to acoustic trauma (noise exposure). Pure-tune noise exposure (16 kHz, 100 dB, 2 h) resulted in much more severe hearing loss in the mutant mice, as evidenced by significant elevations of DPOAE thresholds (S8A Fig), ABR thresholds (S8B Fig) and reductions in ABR P1 amplitudes (S8C Fig). Similarly, noise exposure resulted in higher percentages of OHC loss at 16 and 32 kHz in the mutant mice (S8D and S8E Fig). Together, these results highlight the impact of genetic and environmental influences to the onset and severity of DFNA15 hearing loss.

## The DFNA15 deafness is likely caused by haploinsufficiency of POU4F3

Disease causing mutations in DFNA15 patients have been identified across the entire span of the *POU4F3* gene, including the two POU domains [4, 5]. Whether DFNA15 is caused by dominant negative effects of the mutant POU4F3 proteins or haploinsufficiency of the wildtype POU4F3 protein will be important for understanding the disease mechanisms and treatment approaches. We have recently generated a knockout mouse model with *Pou4f3* coding

region replaced by eGFP after the start codon [20]. At P3, both homozygous *Pou4f3(Δ/Δ)* and *Pou4f3(-/-)* showed extensive degeneration of hair cells, while hair cells of both heterozygous mice were normal (Fig 3A and 3B). This finding confirms the essential role of Pou4f3 in hair cell survival as observed in *Pou4f3(Δ/Δ)* mice (S3B and S7A and S7B Figs). Importantly, both DPOAE and ABR thresholds of the 3 months old *Pou4f3(-/+)* mice exhibit significant elevations at 32 kHz compared to wildtype littermates (Fig 3C–3F). OHCs of the *Pou4f3(-/+)* cochleae were also significantly degenerated at 32 kHz (Fig 3G and 3H). *Pou4f3(-/+)* mice showed reduced ABR P1 amplitudes across the entire cochlear frequencies (Fig 3E and S10A Fig), resembling the 3 months old *Pou4f3(Δ/+)* mice (Fig 1F and S2 Fig).

We then bred the *Pou4f3(-/+)* mice to a mixed genetic background and examined auditory function at 9 months age (S9 Fig). Similarly, 9 months old *Pou4f3(-/+)* mice showed elevations of DPOAE thresholds (S9A Fig), ABR thresholds (S9B Fig) and reductions in ABR P1 amplitudes (S9C and S10B Fig), consistent with the onset of hearing loss (between 6 and 12 months) in *Pou4f3(Δ/+)* mice on the mixed background (S7 Fig). Significant loss of OHCs was also observed in the *Pou4f3(-/+)* mice at multiple cochlear frequencies (S9D and S9E Fig). Together, our findings provide strong support that *POU4F3* haploinsufficiency is the main underlying cause of DFNA15 pathophysiology.

## The Aldh inhibitor DEAB restores auditory function in the DFNA15 mouse model

The cochlear expression of Pou4f3 is regulated by multiple transcription factors and signaling molecules, including Atoh1 [21], Eya1/Six1 [22], Gata3/N-myc/Etv4 [23] and retinoic acid [24]. Among these upstream signals, retinoic acid (RA) signaling was shown to suppress Pou4f3 expression in rat cochlear explants [24] and represents a druggable target by small molecules. We used the aldehyde dehydrogenase (Aldh) inhibitor 4-diethylaminobenzaldehyde (DEAB) [25] and the retinoic acid receptor antagonist BMS195614 [26] to inhibit RA signaling, and found that both DEAB and BMS195614 significantly promoted the expression of *Pou4f3* in wildtype cochlear explants (Fig 4A). Expression of *Tmem237* and *Sra1* were altered in the *Pou4f3(-/+)* cochleae, suggesting these two genes may be regulated by Pou4f3 (S6C Fig); indeed, *Tmem237* but not *Sra1* was upregulated by DEAB (Fig 4B). Interestingly, some of the known Pou4f3 targets, including *Lhx3* and *Caprin1* were upregulated while *Bdnf* was downregulated by DEAB treatment in cochlear explants (Fig 4C).

The levels of RA is sensed and regulated by Cyp26a1, an RA degrading enzyme and a well-recognized transcriptional target of RA [27]. As expected, RA significantly upregulated *Cyp26a1* expression, while DEAB downregulated *Cyp26a1* expression (Fig 4D). Downregulation of Cyp26a1 by DEAB was completely abolished by RA co-treatment (Fig 4D). Although RA treatment itself did not inhibit *Pou4f3* expression, RA co-treatment completely abolished the effect of DEAB on *Pou4f3* expression (Fig 4E), indicating that DEAB regulates *Pou4f3* expression via RA signaling. Regulation of the Pou4f3 downstream target *Lhx3* by DEAB and RA was similar to *Pou4f3* (Fig 4F). In addition, DEAB promoted expression of wildtype Pou4f3 mRNA (Fig 4G) and protein (Fig 4H and 4I) in cochlear explants from the *Pou4f3(Δ/+)* mice. The mutant Pou4f3 protein was not increased by DEAB treatment, possibly due to the effect of long half-life of the mutant Pou4f3 protein [9]. These data suggest that expression of wildtype Pou4f3 and its downstream targets may be rescued by inhibition of RA signaling in the *Pou4f3(Δ/+)* cochleae.

To further examine if RA inhibition could upregulate Pou4f3 expression and restore auditory functions in the DFNA15 mouse model, we administered DEAB to 2–3 months old *Pou4f3(Δ/+)* mice via daily intraperitoneal injections. Similar to cochlear explants, DEAB

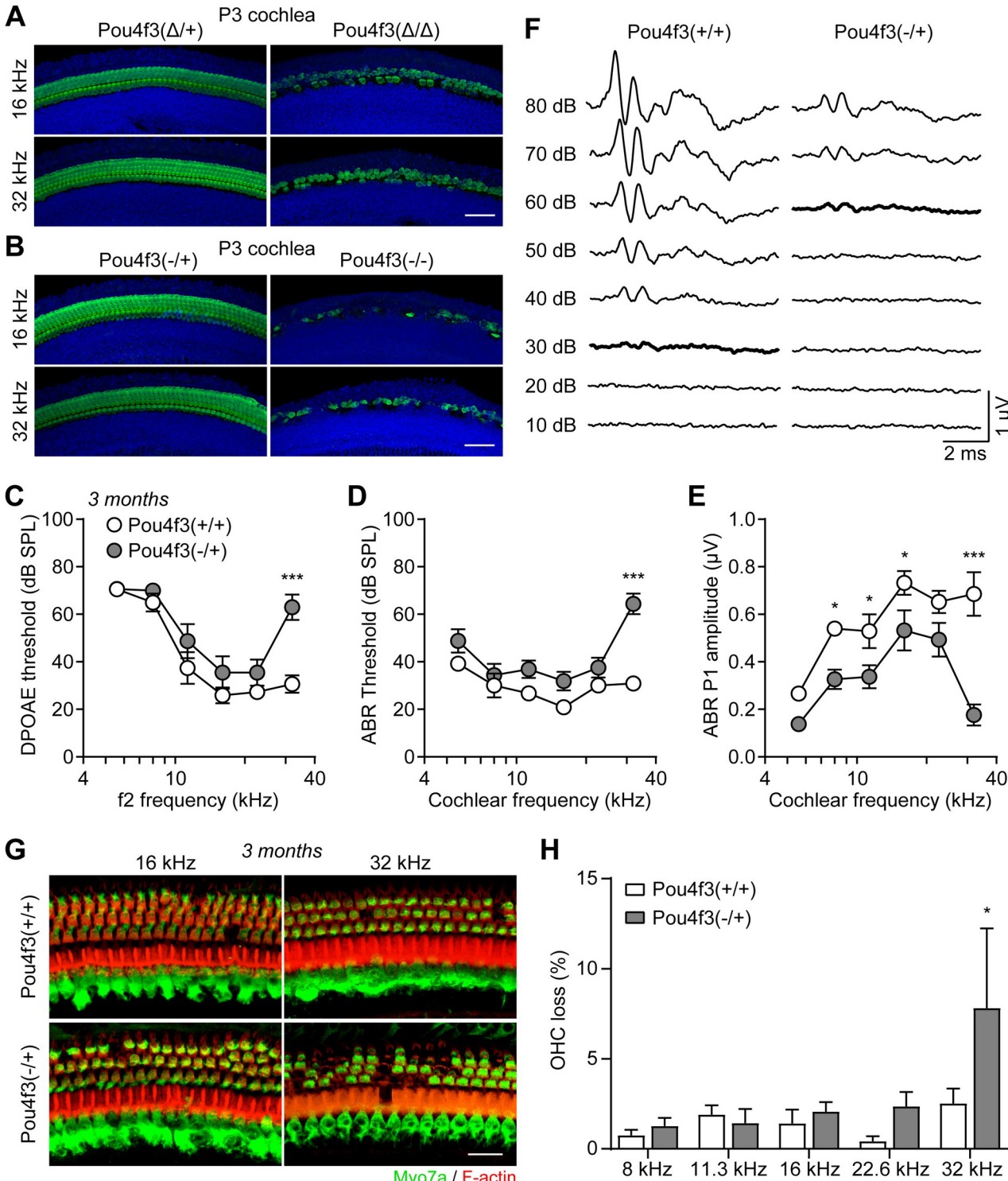

**Fig 3. Heterozygous *Pou4f3* knockout mice display high frequency hearing loss and degeneration of outer hair cells.** (**A-B**) Myo7a immunostaining of cochlear hair cells from (**A**) P3 *Pou4f3(Δ/+)* and *Pou4f3(Δ/Δ)* mice and (**B**) P3 *Pou4f3(-/+)* and *Pou4f3(-/-)* mice. Scale bars were 50 μm. (**C**) DPOAE threshold, (**D**) ABR threshold and (**E**) ABR P1 amplitudes of 3 months old *Pou4f3(+/+)* and *Pou4f3(-/+)* mice. * P < 0.05 and *** P < 0.001 by two-way ANOVA, n = 6–8 mice of each genotype. (**F**) ABR waveforms of 3 months old *Pou4f3(+/+)* and *Pou4f3(-/+)* mice at 32 kHz. Data were mean of n = 6–8 animals. ABR thresholds of each genotype were highlighted. (**G**) Myo7a immunostaining images of the cochlear sensory epithelia from 3 months old *Pou4f3(+/+)* and *Pou4f3(-/+)* mice. Hair cells and F-actin was labelled with Myo7a (green) and Rhodamine-phalloidin (red), respectively. Scale bar was 20 μm. (**H**) Percentage of outer hair cell loss in *Pou4f3(+/+)* and *Pou4f3(-/+)* mice at various cochlear frequencies. * P < 0.05 by two-way ANOVA, n = 8 cochleae of each genotype.

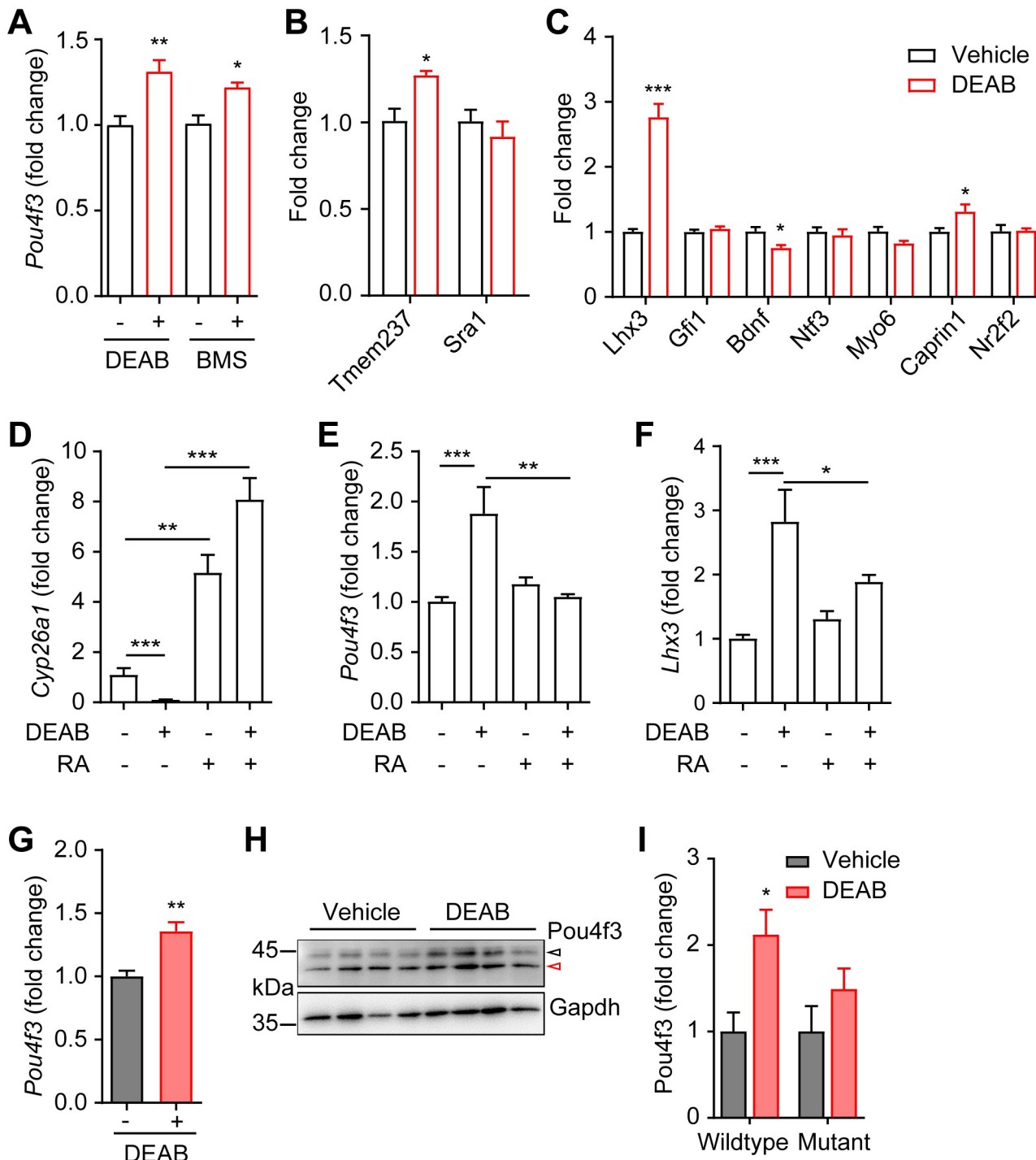

**Fig 4. Inhibition of RA signaling upregulates Pou4f3 expression in cochlear explants.** (**A**) Upregulation of *Pou4f3* mRNA expression by DEAB and BMS195614 in P3 wildtype cochlear explants. DEAB and BMS195614 (BMS) concentrations were 100 and 5 μM, respectively. DMSO was used as control (-). * P < 0.05 and ** P < 0.01 by unpaired student's t-test, n = 13–15 explants for DEAB, n = 4–8 explants for BMS195614. (**B**) Effects of DEAB (red bar) on mRNA expressions of *Tmem237* and *Sra1*. * P < 0.05 by unpaired student's t-test, n = 4 from each treatment. (**C**) Effects of DEAB (red bar) on mRNA expressions of known Pou4f3 targets. * P < 0.05 and *** P < 0.001 by unpaired student's t-test, n = 4 from each treatment. (**D-F**) Effect of DEAB (100 μM) and RA (0.3 μM) treatment on mRNA expressions of (**D**) *Cyp26a1*, (**E**) *Pou4f3* and (**F**) *Lhx3* in P3 wildtype cochlear explants. DMSO was used as control (-). * P < 0.05, ** P < 0.01 and *** P < 0.001 by one-way ANOVA, n = 4 explants for each treatment. (**G**) Upregulation of *Pou4f3* mRNA expression by DEAB in P3 *Pou4f3(Δ/+)* cochlear explants. ** P < 0.01 by unpaired student's t-test, n = 6 explants for each treatment. (**H**) Western blots and (**I**) densitometrical measurements show upregulation of wildtype Pou4f3 protein expression by DEAB in *Pou4f3(Δ/+)* cochlear explants. The black open arrow indicates the wildtype Pou4f3 and red open arrow shows the truncated mutant Pou4f3 protein. * P < 0.05 by unpaired student's t-test, n = 4 explants from each treatment.

treatment *in vivo* also resulted in upregulations of *Pou4f3* and its downstream target *Lhx3* (Fig 5A). However, Cyp26a1 expression remained unchanged, possibly due to complex regulatory mechanisms of *Cyp26a1 in vivo*, a phenotype previously observed with DEAB-treated zebrafish embryos [28].

We then examined the *in vivo* physiological effects of DEAB treatment with DPOAE and ABR tests performed 1 day after last vehicle or DEAB injections. *Pou4f3(Δ/+)* mice showed significant deteriorations in auditory functions over a 3-week period with vehicle treatment, as indicated by elevations in DPOAE threshold at 32 kHz (Fig 5B; grey), ABR threshold at 32 kHz (Fig 5C; grey) and reductions in ABR P1 amplitudes from 8–32 kHz (Fig 5D; grey). Remarkably, the auditory functions of DEAB-treated *Pou4f3(Δ/+)* mice were completely maintained over the 3-week period (Fig 5B–5D; red). We then analyzed the time course of DEAB effects on the auditory function using ABR recordings. Prior to injections, the 2 months old *Pou4f3(Δ/+)* mice in either vehicle or DEAB groups displayed similar ABR thresholds (Fig 5E). The beneficial effects of DEAB were apparent within 2 weeks of treatment at 32 kHz and extended to lower frequencies by 6 and 9 weeks of treatment (Fig 5E). Histological analyses of the cochlear tissue collected after 9-week DEAB treatment also showed significant improvement of OHC survival compared to vehicle-treated *Pou4f3(Δ/+)* mice (Fig 5F and 5G). Together, these results indicate that RA signaling modulates auditory function in the *Pou4f3 (Δ/+)* mice and that small molecule Aldh and RA receptor inhibitors may serve as novel therapeutics for hearing restoration in DFNA15 progressive hearing loss.

## Discussion

In this study, we generated a precision mouse model for human DFNA15 deafness and found that the mutant mice displayed remarkably similar phenotypes as in patients, including impaired DPOAE and progressive hearing loss that starts at high frequencies. This animal model allowed us to reveal the pathological features in DFNA15 cochleae, including fusion, elongation, degeneration of hair cell stereocilia and reduction of hair cell mitochondrial density. Our results also indicate that the onset of DFNA15 deafness is modified by genetic and environment factors and that *POU4F3* haploinsufficiency is the main underlying cause of DFNA15 deafness. Importantly, we identified a small molecule Aldh inhibitor DEAB that promotes Pou4f3 expression in the cochleae and demonstrated its potent effects on rescue of auditory functions in the disease mouse model.

A number of approaches have been taken to identify the disease-causing genes of the DFNA15 deafness, including pedigree and linkage analysis followed by targeted sequencing [3, 10, 29–31], copy number variation (CNV) analyses [32, 33] and more recently whole-exome sequencing [34–39]. All these reports point to POU4F3 as the candidate DFNA15 gene in patients that suffer this autosomal dominant progressive hearing loss. In vitro studies showed that the Pou4f3-8del mutant protein did not negatively affect the transcriptional activities of the wildtype Pou4f3 [9]. Importantly, deletion of the entire POU4F3 gene has been reported in a DFNA15 deafness family [32]. These previous data indicate that DFNA15 is likely caused by haploinsufficiency of the wildtype POU4F3 protein. However, mice either having Pou4f3 gene knocked out, i.e. Pou4f3(-/-) [12, 13] or carrying a spontaneous Pou4f3 mutation, i.e. dreidel, Pou4f3(ddl/ddl) [7] appear to display recessive form of hearing loss, casting doubts on the clinical relevance of these previous animal models and the functional relevance of Pou4f3 haploinsufficiency in DFNA15 pathology. In this study, we found that both Pou4f3(Δ/+) knockin and Pou4f3(-/+) knockout mouse models displayed hearing loss on both C57BL/6J and mixed genetic backgrounds, suggesting that POU4F3 haploinsufficiency is sufficient to cause hearing loss reported in the DFNA15 patients. As the auditory phenotypes of Pou4f3(-/+) knockout

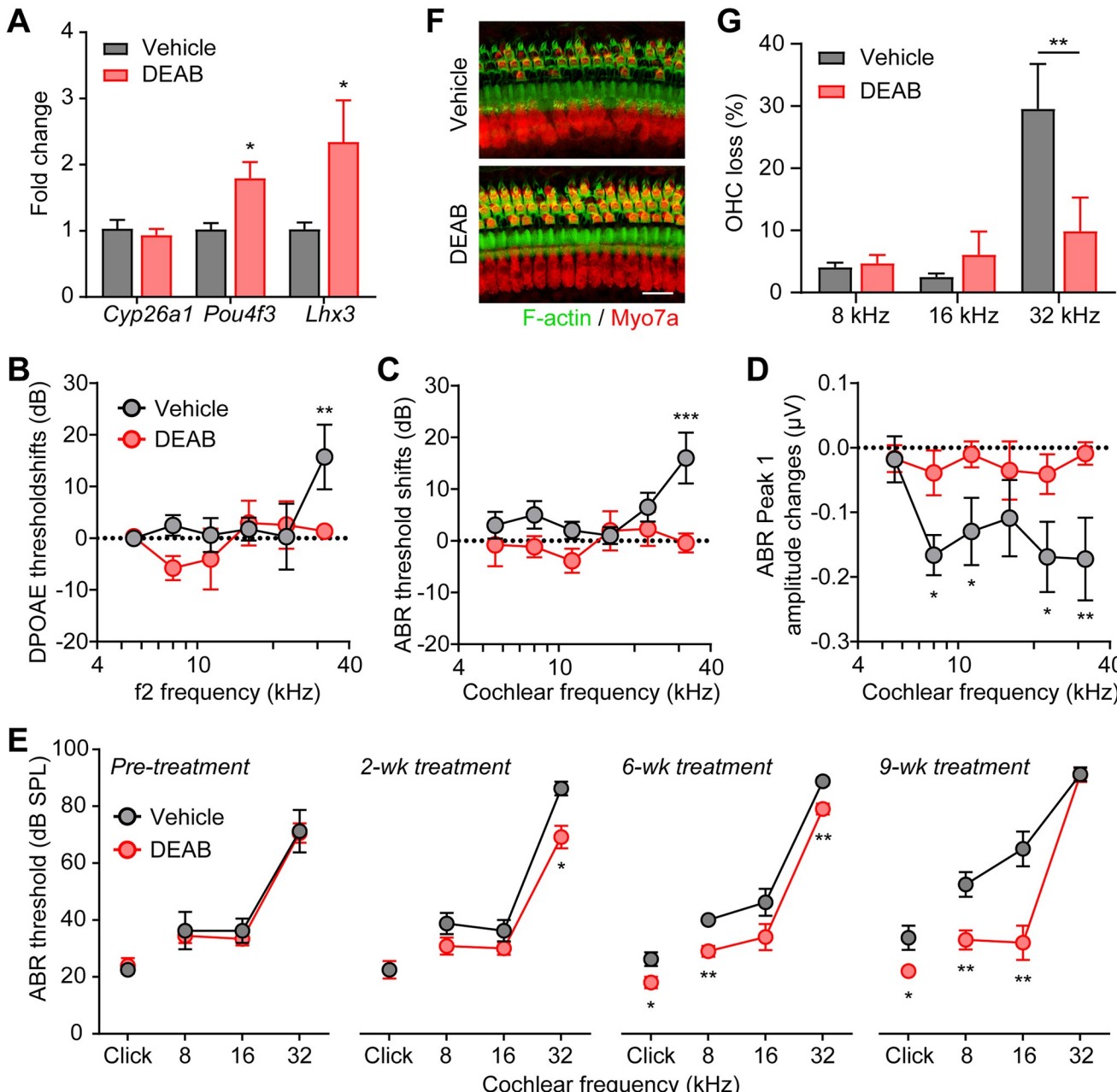

**Fig 5. Inhibition of RA signaling rescues auditory function in *Pou4f3(Δ/+)* mice.** (**A**) Upregulation of *Pou4f3* and *Lhx3* mRNA expression in *Pou4f3(Δ/+)* cochleae by DEAB treatment *in vivo*. DEAB or DMSO vehicle were injected intraperitoneally daily to 3 months old *Pou4f3(Δ/+)* mice for 4 weeks. * $P < 0.05$ by unpaired student's t-test, n = 4–6 mice. (**B**) DPOAE threshold shifts, (**C**) ABR threshold shifts and (**D**) ABR P1 amplitude changes in 3 months old *Pou4f3(Δ/+)* mice treated intraperitoneally daily with either vehicle control or DEAB for 3 weeks. The threshold shifts or amplitude changes were calculated by differences between pre- and 3 weeks post treatment. * $P < 0.05$, ** $P < 0.01$ and *** $P < 0.001$ by two-way ANOVA, n = 10–13 mice of each treatment. (**E**) ABR thresholds of *Pou4f3(Δ/+)* mice treated intraperitoneally daily with either vehicle or DEAB for 2, 6 and 9 weeks. *Pou4f3(Δ/+)* mice were 2 months old prior to treatment. * $P < 0.05$ and ** $P < 0.01$ by unpaired student's t-test, n = 4–6 mice for each treatment. (**F**) Myo7a immunostaining images of the cochlear sensory epithelia (32 kHz region) from *Pou4f3(Δ/+)* mice treated with vehicle or DEAB for 9 weeks. *Pou4f3(Δ/+)* mice were 2 months old prior to treatment. Hair cells and F-actin was labelled with Myo7a (red) and Alexa Fluor 488-Phalloidin (green), respectively. Scale bar was 20 μm. (**G**) Percentage of outer hair cell loss of *Pou4f3(Δ/+)* mice treated with vehicle or DEAB for 9 weeks. ** $P < 0.01$ by unpaired student's t-test, n = 5–10 cochleae for each treatment.

appeared to be milder than the Pou4f3(Δ/+) knockin mice, it remains a possibility that the mutant Pou4f3 may serve a negative role to exacerbate the effects of Pou4f3 haploinsufficiency on hair cell survival and hearing loss.

While the Pou4f3(Δ/+) and Pou4f3(-/+) mice in our study displayed adult-onset progressive hearing loss, no significant auditory phenotype has been observed in heterozygous knockout or dreidel mice at similar ages [7, 40]. This discrepancy could arise from both auditory test methods and genetic background or housing environment of these mutant mice. First, click-evoked ABR recordings are known to be less sensitive in detecting frequency-specific hearing loss [41, 42]. Therefore, it is possible that the previously reported heterozygous mice may have also developed high-frequency hearing loss that was not detected by click-evoked ABR. Second, we show that both genetic and environmental factors exhibit significant contributions to the onset and severity of DFNA15 deafness. Thus, the mouse strain or sub-strain and the housing environment could also contribute to the onset and progression of auditory phenotypes in these heterozygous Pou4f3 mutants. A thorough assessment of the previous heterozygous Pou4f3 mutants at various ages with pure-tone ABR tests may help decipher these phenotypic discrepancies.

POU4F3 is a member of the POU family of transcription factors that regulate a wide array of neuroendocrine developmental pathways [13, 43]. This family is characterized by the presence of a biparitite DNA binding domain known as the POU domain which comprises a POU-homeo domain and a POU-specific domain separated by a linker, both of which are required for sequence-specific DNA binding [44]. A number of Pou4f3 target genes have been previously identified, including *Lhx3* [8], *Gfi1* [7], *Bdnf* [6], *Ntf3* [6], *Myo6* [16], *Caprin1* [17] and *Nr2f2* [18]; however, none of these genes showed apparent alterations in the *Pou4f3(Δ/+)* cochleae examined by RNA-seq or RT-qPCR. One possibility is that changes in expression of these known targets may be subtle and require enrichment of hair cells for their sensitive detection. It is also likely that Pou4f3 may exert its effects via regulation of yet unidentified downstream targets, including those involved in cellular metabolism or biosynthesis based on RNA-seq results. Indeed, *Tmem237*, gene implicated in ciliogenesis [45], and *Sra1*, a ncRNA involved in tumorigenesis [46], were downregulated in the *Pou4f3(Δ/+)* cochleae. Target validation and functional studies of these genes in auditory phenotypes will be a daunting undertaking but remain essential for revealing the molecular basis of DFNA15 deafness.

For the treatment of haploinsufficiency-based dominant form of genetic diseases, an interesting therapeutic approach is to boost the expression or function of the endogenous wildtype genes/proteins, by means of either small molecule-based activation [47] or recently, CRISPR-Cas9 based gene activation [48]. This approach does not rely on the *a priori* knowledge on the downstream targets or effectors of the mutant gene, particularly if the mutant gene is a transcription factor affecting multiple targets responsible for the disease phenotype [49–51]. The two *Pou4f3* mouse models allowed us to confirm that DFNA15 deafness is caused by haploinsufficiency in Pou4f3 expression, which then facilitates the identification and modulation of RA signaling for Pou4f3 upregulation and treatment of DFNA15 deafness.

Retinoic acid (RA) signaling is not only critical for organogenesis and patterning during development, but also important for the maintenance of tissue homeostasis after birth [52]. Therapies based on retinoids to augment RA signaling have been applied to treat various cancers and skin disorders [53]. Alternatively, reducing RA signaling by RORc inhibitors serves as potential therapeutics to combat autoimmune diseases [54], indicating that RA signaling-based therapies are context-dependent. In our study, Aldh inhibitor DEAB administered intraperitoneally daily showed potent effects on Pou4f3 upregulation and rescue of DFNA15 deafness in mice. This is consistent with previous report that DEAB is a potent and safe inhibitor of Aldh *in vivo*, without systemic toxicity even at high doses [55]. In addition, systematic or

local applications of DEAB or another Aldh inhibitor Citral have been used to treat breast cancer and ocular scarring in mouse models [56–58], lending further support to potential applications of Aldh inhibitors in disease treatment. However, as DEAB is short lived *in vivo* [55], development of derivatives or analogues with longer action duration may be advantageous. Lastly, DFNA15 patients suffer from progressive hearing loss, the need to continuously boost Pou4f3 expression for effective treatment remains to be explored and may represent a potential obstacle for drug formulation.

Stereociliary defects, mitochondrial abnormalities or altered cellular metabolism observed in DFNA15 deafness mice are also reported in hearing loss caused by noise, aging and ototoxicity [59–62]. Therefore, upregulation of Pou4f3 by DEAB or other RA antagonists (e.g. BMS195614) may also provide therapeutic potential for treatment of other types of hearing loss. Additionally, as an essential hair cell-specific transcription factor, Pou4f3 also plays instructive roles in hair cell reprogramming [63–65]. Modulation of Pou4f3 expression by DEAB, BMS195614 and other RA signaling inhibitors thus also presents an attractive approach for hair cell regeneration.

## Materials and methods

### Animals

In family H, POU4F3 gene (gene ID:5459) is mutated by an 8-bp deletion, which leads to a shift of coding frame at codon 295 and a premature translation stop at position 299. To introduce the identical mutation into mice, we designed a target vector for knockin (Ki) strategy. The Pou4f3 targeting vector was constructed using BAC (bacterial artificial chromosome)-retrieval methods. In brief, the Pou4f3 locus of mouse (gene ID: 18998) was retrieved from a 129/sv BAC clone (provided by Sanger Institute) by a retrieval vector containing two homologous arms. The 3'-end of exon 2 was replaced by a loxP-Neo-loxP cassette with an 8-nucleotide deletion and an addition of a stop codon created by reversing C to T (C-T). W4 embryonic stem (ES) cells were electroporated with the linearized targeting vector, selected, and then expanded for Southern blot analysis. Chimeric mice were generated by injecting ES cells into C57BL/6J blastocysts followed by transfer into pseudopregnant mice. The germ line transmission of the offspring was identified by genomic PCR and Southern blot. PCR primer pairs were 5'-TCG ACT AGA GCT TGC GGA A-3' and 5'-GAT CTG AAA CCA CCA ACC TC-3'. For genotyping by Southern blot, a 716-bp probe was prepared by PCR using the following primer pairs: forward, 5'-GGG TGT TCA CGA CAA AGT GA-3', and reverse, 5'-CTT GAA ATC CTA GAG GGA AA-3'. Genomic DNA was digested with EcoRV overnight, blotted onto a Nylon membrane and incubated with an $\alpha$-$^{32}$P-labeled probe.

*Pou4f3* knockout mice were generated by replacing the coding sequence after start codon by EGFP [20]. Mice used in this study were either maintained on C57BL/6J background or a mixed background of C57BL/6J and FVBN. For simplicity, unless otherwise specified, results were obtained from mice on C57BL/6J background. All the mice used here were specific pathogen-free (SPF) animals that maintained in standard animal rooms of the National Resource Center for Mutant Mice (NRCMM) of China. All experiments were approved by the Animal Care and Use Committee and carried out in accordance with the animal protocol of Model Animal Research Center of Nanjing University (permit number AP #MZ15 and #WGQ01).

### Cell culture and transfection

HeLa cells were culture in Dulbecco's Modified Eagle Medium (DMEM) supplemented with 10% fetal bovine serum (Gibco, USA). Wildtype mouse Pou4f3 coding sequence and mutant sequence (with 8bp deletion and C-T reversion) were subcloned to pIRES-EGFP plasmid and

transfected to HeLa cells with Lipofectamine 2000 (Life Technologies, USA). Cells were fixed with 4% paraformaldehyde 24 h after transfection.

## Western blot analyses

The intact cochleae were freshly isolated after sacrificing the mice with an overdose of anesthesia and lysed with a lysis buffer containing 2% SDS, 10 mM dithiothreitol, 10% glycerol, a trace amount of Bromophenol Blue and 50 mM Tris HCl, pH 7.4 in 4°C for 1 hour. After homogenization and centrifugation, the protein samples were subjected to sodium dodecyl sulfate-polyacrylamide gel electrophoresis (SDS-PAGE) and transfer to PVDF membrane. The membrane was respectively probed with mouse anti-Pou4f3 (Santa Cruz Biotechnologies, CA, USA) and mouse anti-GAPDH (BioWorld, MN, USA) antibodies followed by incubation with a corresponding secondary antibody. The signals were visualized by incubation with the ECL substrate (Mucyte, China).

## Noise exposure

Both wildtype and mutant mice on a mixed genetic background were subjected to noise exposure at 4 months age. Individual mouse was placed within small cells in a subdivided cage on a rotation station and suspended in a reverberant noise exposure chamber. Noise exposure was executed using LabState software (AniLab Software & Instruments, China) to produce 16 kHz pure-tune noise at 100 dB for 2 h.

## DPOAE and ABR measurements

To assess the hearing function of mice, distortion product otoacoustic emissions (DPOAEs) and auditory brainstem responses (ABRs) were measured as previously described [66, 67]. Briefly, DPOAEs and ABRs were performed on mice anaesthetized with xylazine (20 mg/kg, i.p.) and ketamine (100 mg/kg, i.p.). The DPOAEs in response to two primary tones of frequency f1 and f2 were recorded at $(2 \times f1) - f2$, with f2/f1 = 1.2, and the f2 level 10 dB lower than the f1 level. Sound pressure in the ear canal was amplified and digitally sampled at 4 μs intervals. DPOAE thresholds were identified as the f1 level required to evoke a response at −10 dB SPL. For ABR recordings, needle electrodes were placed under the skin (a) at the dorsal midline between the two ear flaps, (b) behind the right pinna and (c) at the base of the tail (for a ground recording). ABR waveforms were evoked with 5 ms tone pips (0.5 ms rise-fall interval, with a cos2 envelope, at 40/s) delivered to the eardrum at log-spaced frequencies at 5.6, 8, 11.3, 16, 22.6 and 32 kHz. The response was amplified (10,000 ×) and filtered (0.3–3 kHz) with an analog-to-digital board in a PC-based data-acquisition system. Sound level was raised in 5 dB steps from 10 to 80 dB SPL. At each level, 1,024 responses were averaged (with alternating polarity) after "artefact rejection". Both DPOAE and ABR recordings were performed using EPL cochlear function test suite (Mass Eye and Ear, Boston, MA, USA). ABR peak amplitudes were analysed with excel and ABR peak analysis software (Mass Eye and Ear) and presented by averaging data from 60, 70 and 80 dB SPL. Cohorts of animals were also tested with TDT auditory systems. Briefly, the response of click and tone pips of 8, 16 and 32 kHz generated using an evoked generation workstation system III (Tucker Davis Technologies Incorporated, Gainesville, FL, USA) powered by SigGen32 software was averaged (n = 1024) and displayed from 110 dB to 0 dB, decreasing in 5 dB steps.

## Rotarod tests

Vestibular function of 7 months wildtype and mutant mice was performed using a rotarod system (ZB-200, Chengdu Techman Software, China). All the mice were trained at 15 round per minute (rpm) for 240 s each day for three consecutive days before the experiment. On the test day, the mice were tested with three different rotarod test protocols: 1) fixed speed at 12 rpm for 240 s; 2) fixed speed at 20 rpm for 240 s and 3) accelerating speed from 0–44 rpm at 1 rpm/s acceleration and maintained at 44 rpm from 44 to 60 s. All test protocols were carried out for three times with 30–60 m resting time in between. The latencies to drop from the rotarod were recorded and presented as averages of the three tests for each test protocol.

## Plastic sections and wholemount immunofluorescence

Mice were sacrificed with an overdose of anesthesia and then infused with phosphate-buffered solution (PBS). The isolated cochlea was fixed with 4% paraformaldehyde (PFA). For plastic sections, decalcification was performed with 10% (w/v) ethylene diamine tetra-acetic acid (EDTA) for 3 days on a shaker followed by gradient dehydration using ethanol. The dehydrated specimens were penetrated and embedded with MC-Plastic I Kit (MuCyte, China) in 4°C overnight. The embedded blocks were cut and stained by hematoxylin/eosin (H&E).

For immunofluorescence of P10 or adult inner ear, whole mounts of cochleae or utricular sensory epithelia were dissected after decalcification with 10% EDTA overnight. The primary antibodies used were rabbit anti-Myosin VIIa (Myo7a) and mouse anti-Pou4f3 (both from Santa Cruz Biotechnologies, USA). Secondary antibodies were Alexa Fluor 546 or Alexa Flour 488-conjugated goat antibodies (Life Technologies, USA). F-actin was labeled by Alexa Fluor 488- or Rhodamine-conjugated phalloidin (Life Technologies, USA). Samples were examined using a Leica SP5-II (Leica, Germany) or Zeiss LSM880 confocal microscope (Zeiss, Germany). ImageJ software (version 1.46r, NIH, MD) was used for image processing of confocal z-stacks. Percentage of OHC loss was defined as number of OHC missing divided by total OHCs demarcated by F-actin stainings.

## Scanning (SEM) and transmission (TEM) electron microscopy

Mice were fixed by perfusion with PBS containing 2.5% glutaraldehyde following an overdose of anesthesia. The inner ear tissues were isolated and decalcified with 10% (w/v) EDTA for 2 days. The epithelia of organ of Corti were exposed and then fixed with 1% $OsO_4$ in $H_2O$ for 2 hours. For SEM examination, the tissues were dehydrated in an ethanol series and critically point-dried. The dried samples were mounted on stubs, sputter-coated with gold and examined on S-3000 N scanning electron microscope (Hitachi, Tokyo, Japan) at 15 kV. For TEM analyses, samples were dehydrated, infiltrated, and polymerized in araldite. Ultrathin sections (70 nm) were post-stained and examined under a Hitachi-7650 transmission electron microscope (Hitachi) at 70 kV.

## RNA-seq analyses

Total RNA of cochleae from 2 months old *Pou4f3(+/+)* and *Pou4f3(Δ/+)* mice was extracted using RNAiso plus kit (TaKaRa, Japan). Total RNA of each sample was quantified and qualified by Agilent 2100 Bioanalyzer (Agilent Technologies, CA, USA), NanoDrop (Thermo Fisher Scientific, USA) and 1% agarose gel. 1 μg total RNA with RIN value above 6.5 was used for following library preparation. Next generation sequencing library preparations were constructed according to the manufacturer's protocol. The poly(A) mRNA isolation was performed using Poly(A) mRNA Magnetic Isolation Module or rRNA removal Kit. The mRNA fragmentation

and priming was performed using First Strand Synthesis Reaction Buffer and Random Primers. First strand cDNA was synthesized using ProtoScript II Reverse Transcriptase and the second-strand cDNA was synthesized using Second Strand Synthesis Enzyme Mix. The purified doublestranded cDNA by beads was then treated with End Prep Enzyme Mix to repair both ends and add a dAtailing in one reaction, followed by a T-A ligation to add adaptors to both ends. Size selection of Adaptorligated DNA was then performed using beads, and fragments of ~420 bp (with the approximate insert size of 300 bp) were recovered. Each sample was then amplified by PCR for 13 cycles using P5 and P7 primers, with both primers carrying sequences which can anneal with flow cell to perform bridge PCR and P7 primer carrying a six-base index allowing for multiplexing. The PCR products were cleaned up using beads, validated using a Qsep100 (Bioptic, Taiwan, China), and quantified by Qubit3.0 Fluorometer (Invitrogen, CA, USA). Then libraries with different indices were multiplexed and loaded on an Illumina HiSeq instrument according to manufacturer's instructions (Illumina, San Diego, CA, USA). Sequencing was carried out using a 2x150bp paired-end (PE) configuration; image analysis and base calling were conducted by the HiSeq Control Software (HCS) + OLB + GAPipeline-1.6 (Illumina) on the HiSeq instrument. The sequences were processed and analyzed by GENEWIZ. Differential expression analysis used the DESeq2 Bioconductor package, a model based on the negative binomial distribution. The estimates of dispersion and logarithmic fold changes incorporate data-driven prior distributions, Padj of genes were set < 0.05 to detect differential expressed ones. Gene ontology (GO) analyses of biological processes enriched in the differentially regulated genes were performed at geneontology.org [68, 69]. The RNA-seq data were available at GEO database with accession number GSE139145.

## Cochlear explant culture

Cochlear sensory epithelia were isolated from 3 days old wildtype or *Pou4f3(Δ/+)* mice and transferred to HBSS. Cochlear explants were cultured on 35mm dish coated with collagen I (BD Biosciences, USA), in serum-free 1:1 mixture of DMEM and F12, supplemented with Glutamax, N2, B27 supplements and Penicillin G (all from Gibco, USA). After 24 h, 100 μM 4-Diethylaminobenzaldehyde (DEAB, Sigma, USA), 5 μM BMS195614 (Tocris Bioscience, UK) or 0.3 μM all-trans retinoic acid (RA, Sigma, USA) or DMSO vehicle control were added to the culture medium for 5 days with fresh drug-containing media replaced every 2 days.

## Quantitative RT-PCR (RT-qPCR)

Cultured cochlear explants or isolated whole cochleae were homogenized and total RNA was extracted from tissues with the RNAiso Plus kit (TaKaRa, Japan). The quantity and purity of RNA samples were checked by determining absorbance at 260/280 nm by using ND-1000 spectrophotometer (Thermo Scientific, DE, USA). Reverse transcription reactions were performed by using the HiScript Q RT Super Mix (Vazyme, China) and qPCR was performed with a SYBR Premix Ex Taq kit (TaKaRa, Japan) on a Step One Plus Real-time PCR System (ABI Biosystems, USA). QPCR reactions were performed with the following primers: GAPDH; 5'-ACC ACG AGA AAT ATG ACA ACT CAC-3' and 5'-CCA AAG TTG TCA TGG ATG ACC-3'; Pou4f3; 5'-CCT ATT TCG CCA TCC AGC CA-3' and 5'-TTA CAG AAC CAG ACC CTC ACC A-3'; Cyp26a1; 5'-GCA ATC AAG ACA ACA AGT TAG ACA T-3' and 5'-CAA CCC GAA ACC CTC CTG-3'; Lhx3; 5'-GCA GTT CCA AGT CCG ACA A-3' and 5'-ACG GCT CAT CAG TGA AGG A-3'; Gfi1; 5'- CGC AGG TTA TCA GAG TAA GGA-3' and 5'-CTT GGA AGC ACA GAA CAC AG-3'; Bdnf; 5'-GTG TGT GAC AGT ATT AGC GAG TGG-3' and 5'-GAT ACC GGG ACT TTC TCT AGG AC-3'; Ntf3; 5'-GCC CCC TCC CTT ATA CCT AAT G-3' and 5'-CAT AGC GTT TCC TCC GTG GT-3'; Myo6; 5'-GAC AGC

AGC GTT TCT TCC-3' and 5'-AAT CCA GGG TCC GTC AAA-3'; Caprin1; 5'-GAG CCA GCG GAA GAA TAC AC-3' and 5'-TCA ACT GTC CAC TCA TCC ACT T-3'; Nr2f2; 5'-GGA TCT TCC AAG AGC AAG TG-3' and 5'-AGG CAT CTG AGG TGA ACA-3'; Tmem237; 5'-GAC CGT TCC GAG TTG ATA A-3' and 5'-AGA GAA CAG ACC GAC CAT-3'; Sra1; 5'-ATC CAC CTC CTT CAA GTA-3' and 5'-CGT CTT CTA TCA GAG TTT CA-3'.

## DEAB injections

Both female and male *Pou4f3(Δ/+)* mice aged 2–3 months were used for drug injections. 4-Diethylaminobenzaldehyde (DEAB, Sigma) was dissolved in either dimethyl sulfoxide (DMSO) or ethanol solvents to 200 mg/ml. DEAB stock was further diluted with corn oil at 1:10 (v/v). The DEAB/solvent/corn oil mix was injected intraperitoneally to *Pou4f3(Δ/+)* mice at 100 mg/kg/day of DEAB daily for 2, 3, 6, or 9 weeks. Same volume of solvent/corn oil mix was injected to *Pou4f3(Δ/+)* mice as vehicle controls.

## Statistical analysis

The sample numbers were stated in the figure legends. Data were presented as the mean ± SEM. Differences between groups were determined by unpaired student's t-test, one-way or two-way ANOVA as indicated in the figure legends. All the numerical data were presented in S1 Data.

## Supporting information

**S1 Fig. Generation and characterization of the *Pou4f3(Δ/+)* mice.** (**A**) Schematic representation of the *Pou4f3* knockin strategy using *Pou4f3* gene with the 8 bp deletion and C-T reversion. (**B**) Nucleotide and amino acid sequences of wildtype and mutant *Pou4f3*. Red boxes indicate the 8bp deletion identified in DFNA15 patients. C-T reversion was applied to generate the premature stop codon in the mouse version of mutant *Pou4f3*. (**C**) DNA isolated from chimeric and wild-type mice tails was digested with EcoRV and analyzed by Southern blot. The wild-type and mutant alleles yield 6.7 kilobase (kb) and 8.6kb fragments, respectively. (**D**) PCR genotyping for the *Pou4f3(Δ/+)* mice. A band of 542bp size can be detected in the genomic DNA from the *Pou4f3(Δ/+)* mice. (**E**) Western blot analysis for Pou4f3 protein from cochlear sensory epithelia of P3 mice. The black open arrow indicates the wildtype Pou4f3 in both control and *Pou4f3(Δ/+)* mice while red open arrow shows the truncated Pou4f3 protein. (**F**) Localization of the wildtype and mutant mouse Pou4f3 in HeLa cells. Wildtype of mutant Pou4f3 were cloned to pIRES-eGFP plasmid and expressed in HeLa cells. eGFP was localized to both cytoplasm and nucleus. Scale bar was 10 μm. (**G**) Expression and localization of Pou4f3 protein in outer hair cells of P10 cochleae from *Pou4f3(+/+)* and *Pou4f3(Δ/+)* mice. Arrow heads indicate cytoplasmic localization of the mutant Pou4f3. Scale bar was 20 μm. (TIF)

**S2 Fig. *Pou4f3(Δ/+)* mice display age- and tonotopy-dependent reductions in ABR P1 amplitudes.** (**A-F**) ABR P1 amplitude growth curves of *Pou4f3(+/+)* and *Pou4f3(Δ/+)* mice at 2, 3 and 7 months of age. **A**, 5.6 kHz; **B**, 8 kHz; **C**, 11.3 kHz; **D**, 16 kHz; **E**, 22.6 kHz; **F**, 32 kHz. * $P < 0.05$, ** $P < 0.01$ and *** $P < 0.001$ by two-way ANOVA. (TIF)

**S3 Fig. *Pou4f3(Δ/Δ)* mice are profoundly deaf and lose all hair cells by 1 month age.** (**A**) ABR thresholds for click and pure tones (8, 16 and 32 kHz) of 1-month old *Pou4f3(+/+)* (n = 3), *Pou4f3(Δ/+)* (n = 5) and *Pou4f3(Δ/Δ)* (n = 2) mice. The two *Pou4f3(Δ/Δ)* mice were completely deaf without evocable ABR responses. (**B**) F-actin labelling and Myo7a

immunostaining of the apical cochlear turn from 1 month old *Pou4f3(+/+)*, *Pou4f3(Δ/+)* and *Pou4f3(Δ/Δ)* mice. Both OHCs and IHCs were completely lost in the sensory epithelia of *Pou4f3(Δ/Δ)* mice. Similar result was observed at the basal turn. Scale bars: 50 μm.
(TIF)

**S4 Fig.** *Pou4f3(Δ/+)* **mice do not display vestibular dysfunction by 7 months age.** (**A**) Schematic representation of the rotarod testing protocols. (**B**) The time to fall from the rotarod of *Pou4f3(+/+)* and *Pou4f3(Δ/+)* mice. No significant difference was observed with all 3 testing protocols. n = 11–13 mice of each genotype. (**C**) DPOAE tests of the *Pou4f3(+/+)* and *Pou4f3 (Δ/+)* mice. ** P < 0.01 by two-way ANOVA, n = 10 mice of each genotype. (**D**) Myo7a and Pou4f3 immunofluorescence images showing the entire utricular sensory epithelium from *Pou4f3(+/+)* or *Pou4f3(Δ/+)* mice. Squares represent high magnification samplings of extra-striolar and striolar areas. Scale bar was 100 μm. (**E-F**) Density of utricular hair cells in extra-striolar area (**E**) and striolar area (**F**) in *Pou4f3(+/+)* or *Pou4f3(Δ/+)* mice. (**G**) The surface areas of utricular sensory epithelia in *Pou4f3(+/+)* or *Pou4f3(Δ/+)* mice. * P < 0.05 by unpaired student's t-test, n = 12–13 utricles of each genotype.
(TIF)

**S5 Fig.** *Pou4f3(Δ/+)* **mice display OHC degeneration at cochlear bases.** (**A, C**) Myo7a immunostaining images of the cochlear sensory epithelia from (**A**) 3 months and (**C**) 7 months old *Pou4f3(+/+)* and *Pou4f3(Δ/+)* mice. Hair cells and F-actin was labelled with Myo7a (green) and Rhodamine-phalloidin (red), respectively. Scale bar was 20 μm. (**B, D**) Percentage of outer hair cell loss in (**B**) 3 months and (**D**) 7 months old *Pou4f3(+/+)* and *Pou4f3(Δ/+)* mice. * P < 0.05 and ** P < 0.01 by two-way ANOVA, n = 3–4 cochleae of each genotype.
(TIF)

**S6 Fig. Changes in cochlear gene expression of 2 months old** *Pou4f3(Δ/+)* **mice.** (**A**) Gene expression analyses of Pou4f3 and its known downstream target genes by RT-qPCR. (**B**) Top 20 gene ontology (GO) processes of differentially expressed genes in *Pou4f3(Δ/+)* cochleae. Metabolic processes were highlighted in red. Padj, adjusted P value. (**C**) RT-qPCR validations of selected genes identified from the RNA-seq experiment. * P < 0.05 and *** P < 0.001 by unpaired student's t-test, n = 5 cochleae of each genotype.
(TIF)

**S7 Fig.** *Pou4f3(Δ/+)* **mice display late-onset progressive hearing loss on a mixed genetic background.** (**A**) DPOAE and (**B**) ABR thresholds of 3-week old *Pou4f3(+/+)* (n = 18), *Pou4f3 (Δ/+)* (n = 21) and *Pou4f3(Δ/Δ)* (n = 4) mice. *Pou4f3(Δ/Δ)* mice were completely deaf without evocable ABR responses. *** P < 0.001 by two-way ANOVA. (**C-E**) 3 months (n = 6–10), (**F-H**) 6 months (n = 6) and (**I-K**) 12 months (n = 13–28) old wildtype *Pou4f3(+/+)* and mutant *Pou4f3(Δ/+)* mice were tested with DPOAE and ABR. Mice were maintained on a mixed background of C57BL/6J and FVBN. (**C, F, I**) DPOAE thresholds; (**D, G, J**) ABR thresholds; (**E, H, K**) ABR peak 1 (P1) amplitudes. * P < 0.05, ** P < 0.01 and *** P < 0.001 by two-way ANOVA. (**L**) Myo7a immunostaining images of the cochlear sensory epithelia from 12 months old wildtype and mutant mice. Hair cells and F-actin was labelled with Myo7a (green) and Rhodamine-phalloidin (red), respectively. Scale bar was 20 μm. (**M**) Percentage of outer hair cell loss in wildtype and mutant mice. ** P < 0.01 by two-way ANOVA, n = 3–4 cochleae of each genotype.
(TIF)

**S8 Fig.** *Pou4f3(Δ/+)* **mice are more susceptible to noise-induced hearing loss on a mixed genetic background.** (**A**) DPOAE threshold (n = 12–16), (**B**) ABR threshold (n = 6–8) and (**C**)

ABR P1 amplitudes (n = 6–8) of 4 months old *Pou4f3(+/+)* and *Pou4f3(Δ/+)* mice 10 days after noise exposure. Mice were maintained on a mixed background of C57BL/6J and FVBN. Symbol of speaker and vertical line indicate pure-tone noise exposure at 16 kHz, 100 dB for 2 h. * $P < 0.05$, ** $P < 0.01$ and *** $P < 0.001$ by two-way ANOVA. (**D**) Myo7a immunostaining images of the cochlear sensory epithelia from wildtype and mutant mice 10 days after noise exposure. Hair cells and F-actin was labelled with Myo7a (green) and Rhodamine-phalloidin (red), respectively. Scale bar was 20 μm. (**E**) Percentage of outer hair cell loss in wildtype and mutant mice 10 days after noise exposure. ** $P < 0.01$ by two-way ANOVA, n = 4 cochleae of each genotype.
(TIF)

**S9 Fig. Hearing loss and OHC degeneration of 9 months old *Pou4f3(-/+)* mice on a mixed genetic background.** (**A**) DPOAE threshold, (**B**) ABR threshold and (**C**) ABR P1 amplitudes of 9 months old *Pou4f3(+/+)* and *Pou4f3(-/+)* mice. Mice were maintained on a mixed background of C57BL/6J and FVBN. * $P < 0.05$, ** $P < 0.05$ and *** $P < 0.001$ by two-way ANOVA, n = 4–14 mice of each genotype. (**D**) Myo7a immunostaining images of the cochlear sensory epithelia from 9 months old *Pou4f3(+/+)* and *Pou4f3(-/+)* mice. Hair cells and F-actin was labelled with Myo7a (green) and Rhodamine-phalloidin (red), respectively. Scale bar was 20 μm. (**E**) Percentage of outer hair cell loss in *Pou4f3(+/+)* and *Pou4f3(-/+)* mice at various cochlear frequencies. * $P < 0.05$ and *** $P < 0.001$ by two-way ANOVA, n = 6–10 cochleae of each genotype.
(TIF)

**S10 Fig. Reductions of ABR P1 amplitudes in 3 and 9 months old *Pou4f3(-/+)* mice.** (**A-B**) ABR P1 amplitude growth curves of (**A**) 3 months old *Pou4f3(+/+)* and *Pou4f3(-/+)* mice on C57BL/6J background (n = 6–8) and (**B**) 9 months old *Pou4f3(+/+)* and *Pou4f3(-/+)* mice on a mixed background (n = 4–14), at 5.6, 8, 11.3, 16, 22.6 and 32 kHz. * $P < 0.05$, ** $P < 0.01$ and *** $P < 0.001$ by two-way ANOVA.
(TIF)

**S1 Table. Vestibular phenotypes in DFNA15 patients.**
(DOCX)

**S2 Table. List of differentially expressed genes in *Pou4f3(Δ/+)* cochleae.**
(XLS)

**S1 Data. Raw numerical data of all the figures.**
(XLSX)

## Acknowledgments

We thank Ziyi He (School of Life Sciences, Nanjing Agricultural University) for assistance with scanning electron microscopy analysis.

## Author Contributions

**Conceptualization:** Sihao Gong, Xiao-Yun Qian, Huaqun Chen, Xiang Gao, Jian Zuo, Min-Sheng Zhu, Xia Gao, Guoqiang Wan.

**Data curation:** Guang-Jie Zhu, Sihao Gong, Deng-Bin Ma, Tao Tao, Wei-Qi He, Linqing Zhang, Han Zhou, Guoqiang Wan.

**Formal analysis:** Guang-Jie Zhu, Sihao Gong, Deng-Bin Ma, Tao Tao, Wei-Qi He, Linqing Zhang, Fang Wang, Pei Wang, Min-Sheng Zhu, Guoqiang Wan.

**Funding acquisition:** Jian Zuo, Min-Sheng Zhu, Xia Gao, Guoqiang Wan.

**Investigation:** Guang-Jie Zhu, Sihao Gong, Tao Tao, Wei-Qi He, Linqing Zhang, Fang Wang, Han Zhou, Chi Fan, Pei Wang, Xin Chen, Wei Zhao, Jie Sun, Ye Wang, Guoqiang Wan.

**Methodology:** Guang-Jie Zhu, Sihao Gong, Deng-Bin Ma, Tao Tao, Wei-Qi He, Linqing Zhang, Fang Wang.

**Resources:** Xiang Gao.

**Supervision:** Jian Zuo, Min-Sheng Zhu, Xia Gao, Guoqiang Wan.

**Validation:** Guang-Jie Zhu, Tao Tao, Guoqiang Wan.

**Visualization:** Guoqiang Wan.

**Writing – original draft:** Guang-Jie Zhu, Sihao Gong, Jian Zuo, Min-Sheng Zhu, Xia Gao, Guoqiang Wan.

**Writing – review & editing:** Jian Zuo, Min-Sheng Zhu, Xia Gao, Guoqiang Wan.

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
