## [Decision Letter · Decision Letter 0]

25 May 2020

Dear Dr Wan

Thank you very much for submitting your Research Article entitled 'Aldh inhibitor restores auditory function in a mouse model of human deafness' to PLOS Genetics. Your manuscript was fully evaluated at the editorial level and by independent peer reviewers. The reviewers appreciated the attention to an important problem, but raised some substantial concerns about the current manuscript. Based on the reviews, we will not be able to accept this version of the manuscript, but we would be willing to review again a much-revised version. We cannot, of course, promise publication at that time.

If you decide to revise the manuscript for further consideration at PLOS Genetics, please aim to resubmit within the next 60 days, unless it will take extra time to address the concerns of the reviewers, in which case we would appreciate an expected resubmission date by email to plosgenetics@plos.org.

[LINK]

We are sorry that we cannot be more positive about your manuscript at this stage. Please do not hesitate to contact us if you have any concerns or questions.

Yours sincerely,

Lisa Cunningham, Ph.D.

Guest Editor

PLOS Genetics

Gregory Barsh

Editor-in-Chief

PLOS Genetics

Reviewer's Responses to Questions

**Comments to the Authors:**

Reviewer #1: Mutations in Pou4f3 are associated with autosomal dominant non-syndromic deafness, DFNA15. The manuscript from Zhu et al. reports finding from a study on two mouse models of DFNA15 used to determine how the pathology occurs and to assess novel therapeutic venues.

Zhu et al. generated a knock-in mouse model mimicking a mutation identified in and Israeli family. This mutation leads to expression of truncated Pou4f3 protein and progressive high to mid frequency hearing loss in heterozygous mice. This loss is associated with stereocilia malformation, mitochondria vacuolization and progressive hair cell loss.

To determine if this pathology is associated with a dominant negative effect of the truncated protein or haplo-insufficiency, the same study is performed in a knock-out model of Pou4f3. Similar outcomes are observed in heterozygous KO mice, suggesting that DFNA15 is associated with haplo-insufficiency.

To explore therapeutic venues, the authors assess one of the pathways that may control Pou4f3 expression: retinoic acid has been shown to suppress Pou4F3. They report modulation of Pou4f3 and recovery of hair cells in vitro (P3 treatment) and hair cells and auditory function up to 9 weeks after 2 to 3 months of age, with daily intraperitoneal injections of DEAB (aldh inhibitor) in knock-in mice.

This work reports important finding that relate to the physiopathology of DFNA15 and provides important results that support the use of retinoic acid inhibitors to regulate pou4f3 expression in heterozygous mutants. The manuscript is clearly written, and the figures well laid out. A few major points highlighted below remain to be addressed.

Major:

- The comparison of the KI het phenotype with the KO het phenotype is meant to confirm that the progression of hearing loss in DFNA15 is associated with haploinsufficiency in the KI het mice. ABR, DPOAEs and ABR peak 1 amplitude are only shown at one time point (3 months of age) for the het KO mice. It would be good to show all three time points or at least the 3- and 7-months data which show, by 7 months of age in the het KI mice, further hearing loss in the mid frequencies.

- DFNA15 is associated with hearing and balance deficits in patients. Pou4f3 knockout mouse models have been reported to exhibit vestibular dysfunction and display circling behaviors. This aspect, not discussed in the study, is of utmost importance particularly if treatment with adhl inhibitor can also improve balance behavior.

- Keithley et al. (Hear Res. 1999 Aug;134(1-2):71-6) have reported that hearing of heterozygote Pou4f3

knockout mice is indistinguishable from that of wild-type animals up to 24 months ruling out haploinsufficiency as a cause for deafness in DFNA15. This result (not mentioned) is in contrast with the report of progressive hearing loss in both mouse models assessed in this current manuscript. Can the author discuss this discrepancy?

Minor:

- Supplemental Figure 2: Indicate, on the figure, frequencies assessed for each panels.

- Line 156: Talking about “progression in hearing loss” is confusing here as it is meant to describe hearing improvement.

Reviewer #2: Zhu et al report on their study characterizing two mouse models of DFNA15. One mouse line has a truncating mutation introduced into the Pou4f3 gene identical to a mutation found in humans that causes autosomal dominant non-syndromic deafness. The other model is knock-out of Pou4f3, and heterozygous mice also have hearing loss, demonstrating the DFNA15 results from haploinsufficiency. The authors go on to show that inhibiting retinoic acid signaling by aldehyde dehydrogenase (Aldh) or RA receptor inhibitors stimulate Pou4f3 expression and prevents hearing loss in the mice. The work shows that DFNA15 results from haploinsufficiency of Pou4f3 and that stimulating Pou4f3 pharmacologically can be therapeutic in mice. This study provides some valuable insights into the Pou4f3 mechanism of hearing loss and generally, the data are convincing. However, there are some issues with the reporting and the comprehensiveness of the datasets that raise some concerns. The authors should take more care in describing the ages that mice were analyzed and the rationale for the selection and also discuss discrepancies and shortcomings. There are also a number of analyses that could be easily performed that would increase the impact of the results. These points are detailed below.

Additional details on the nature of the mice should be included. For the knockin mice, the description and figure S1 leave some ambiguity as to the exact nature of the sequence in the mice. It would be helpful to also include the amino acid sequence of the truncated protein or a cartoon indicating the differences between WT and truncated form.

For the knockout mice there is almost no description of how the mouse was generated although it appears that these mice have been previously described in another publication by the authors and this should be cited in the methods (line 228).

Data not shown (line 186) should be provided as a supplemental figure. This particular analysis will be of interest considering that it shows lack of consistency with previous published data on known targets of Pou4f3.

The authors should provide a rationale for why RNA-Seq analysis was done on 2 month old mice, which do not yet have detectable hearing deficits. In addition, a file with the raw data used to compile S1 and S2 tables should be included.

Validation of gene expression changes reported in the knockin mice could be done in the explants tested in Fig. 4 with the DEAB, BMS and RA, would strengthen the mechanistic links.

It would be valuable to analyze the level of Pou4f3 and Cyp26a1 in the cochlea of the treated mice shown in Fig. 4G-J) to demonstrate treatment effect on expected target.

Why was the 16 kHz region measures in Fig. 4L? The mice, as analyzed in Fig. 2, do not appear to have OHC loss in that region at 4 months (Fig. 2) which is the age that the treated mice would be (4- 5 months if treated at 2-3 months as indicated in methods).

A discussion of the therapeutic potential of inhibiting RA signaling as a therapeutic strategy and in particular of DEAB and BMS in particular would be informative.

Numerous issues with grammar and writing style- care should be taken in editing.

Line 25: “…is one of the most common autosomal dominant non-syndromic deafness.” Should be “is one of the most common causes of/forms of autosomal dominant…”

Line 217: DNA/RNA sequence should be referred to by 5’ and 3’ end rather than by C-terminus

Figure S1: revesion should be reversion

Please define all abbreviations used throughout, e.g. FC, padj, dB, SPL, etc.

Reviewer #3: Zhu et al have created a knockin mouse model of the first DFNA15 mutation found in humans and demonstrate that the heterozygous mice have a progressive hearing loss similar to the phenotype in affected humans who have a single copy of the dominant mutation. Taken together with the identification similar hearing phenotype in a full knockout Pou4f3 mouse model they suggest this provides confirmation that the mechanism underlying the progressive hearing loss in humans is haploinsufficiency rather than due to a dominant negative action of the mutation. Previous data had suggested that mice heterozygous for Pou4f3 knockout did not have a hearing phenotype. Furthermore, they then use IP delivery of an inhibitor of retinoic acid signalling to ameliorate the progression of the hearing loss in this mouse model suggesting that this may be an approach of therapeutic benefit in humans with DFNA15.

On the whole the dataset is persuasive and raises the exciting possibility of a drug approach to treating an adult onset deafness. However, there are several areas where the manuscript has limitations and should be improved, none of these are major weaknesses that would prevent publication but several do need addressing.

1. Please explain more clearly the nature of the mutation created in the knock-in and the reason for having to introduce a STOP codon. I am guessing this is because the mouse sequence differs to the human one at that base and therefore the 8bp deletion would not create a STOP codon in the mouse sequence but this should be made clear. Does the knock-in fully recapitulate the human mutation with the introduction of the same 4 novel amino acids? This information should be added to FigS1.

2. The manuscript fails to summarise the previous knowledge of DFNA15 or discuss how these findings change our understanding of the effect of this mutation in 2 important respects:

a) There are several previous POU4F3 knock-out mouse models and a spontaneous Dreidel mutant with a 2bp deletion. None of these mice are reported to have a hearing deficit in the heterozygote, although only one has been aged out to 24 months and assessed PMID: 10452377. This is not mentioned in the manuscript. Can the authors suggest a reason for the difference? Are the strain backgrounds the same?

b) The second omission is to the work of Weiss et al which suggest that this 8pb del causes a mutant protein which has a longer half-life than the wildtype protein and is localised to the cytoplasm in a cell line model instead of the nucleus. They suggested that this aberrant protein could cause cell death and that this might underlie the hair cell loss in DFNA15 patients. Do the authors consider that this is still a possibility and did they test for where the mutant protein is localised in the knock-in?

3. Given that previous knock-outs of Pou4f3 have not reported a het phenotype, it is particularly important to be sure that the data from this new model is definitive. Currently, hearing data for only 1 age point is presented for this mouse, at 3 months, when the hearing deficit is modest. Addition of data for 6-7 months would strengthen the case for a het phenotype.

4. Beginning at line 138, the authors suggest that Pou4f3 is regulated by retinoic acid signalling and has been shown to be supressed by retinoic acid in rat cochlear explants, citing ref 17 for both statements. Ref 17 concerns GATA3 and Prestin expression and does not mention Pou4f3. I have been unable to find another paper describing Pou4f3 being regulated by retinoic acid. Could the authors explain on what they based their rationale for using retinoic acid inhibitors?

5. The authors document several morphological deficits in the DFNA15 model, including OHC loss, stereocilia and mitochondrial deficits but there is an absence of functional assays. Have the authors considered using FM-143 or mitochondrial assays to test functionality on hair cells?

6. Are their balance defects in the 2 mouse models?

7. Have the RNAseq data been validated by qPCR for any of the genes?

8. Currently the evidence suggest that DEAB can prevent or slow progression of the hearing loss but it would be interesting to know whether there is an improvement to pre-treatment hearing levels at different ages, suggesting that damaged stereocilia and mitochondrial function can be repaired by Pou4f3 induction. Is this something the authors have assessed?

9. In the discussion the authors suggest this could form the basis of a therapeutic approach to DFNA15 patients. It would be good to temper this statement with some acknowledgement of the difficulties and barriers to be overcome before this is a viable option.

Other points:

Fig S1 reversion is misspelt.

Define abbreviations, FDA, H&E …

Fig 2 % OHC loss- please define what the percentage is of?

Fig2 and Fig3 for clarity please put the ages on the charts rather than the legends.

Fig 4A-D. What is the assay? qPCR? What is FC?

Fig 4E&F. Please quantify the mutant and wildtype protein separately, there seems to be a difference in effect.

Fig 4G-I. Please clarify whether the ABRs were immediately after 3 weeks’ treatment or 3 weeks after the treatment finished.

The authors should consider depositing their RNAseq data in the G-EAR portal.

**Have all data underlying the figures and results presented in the manuscript been provided?**

Reviewer #1: No: Data are available but not provided as supported information

Reviewer #2: Yes

Reviewer #3: Yes

PLOS authors have the option to publish the peer review history of their article (what does this mean?). If published, this will include your full peer review and any attached files.

Reviewer #1: No

Reviewer #2: No

Reviewer #3: No

---

## [Decision Letter · Decision Letter 1]

5 Aug 2020

Dear Dr Wan,

Thank you very much for submitting your Research Article entitled 'Aldh inhibitor restores auditory function in a mouse model of human deafness' to PLOS Genetics. Your manuscript was fully evaluated at the editorial level and by independent peer reviewers. The reviewers appreciated the attention to an important topic but identified some aspects of the manuscript that should be improved.

We therefore ask you to modify the manuscript according to the review recommendations before we can consider your manuscript for acceptance. Your revisions should address the specific points made by each reviewer.

[LINK]

Yours sincerely,

Lisa Cunningham, Ph.D.

Guest Editor

PLOS Genetics

Gregory Barsh

Editor-in-Chief

PLOS Genetics

The Reviewers and Editors appreciate the thoughtful responses and additional data provided by the Authors. The manuscript is substantially improved, and there is enthusiasm for the paper overall. The Reviewers require minor clarifications that will further improve the paper. Please note that the comments of one of the Reviewers are provided as an attachment, while those of the other two are appended below. We look forward to the revised manuscript.

Reviewer's Responses to Questions

**Comments to the Authors:**

Reviewer #1: The authors have substantially revised the manuscript and included new data that address this reviewer's concerns. In particular they have now included new data that highlight effect of genetic background and environmental insults to hearing loss progression associated with DFNA15. They have also now included rotarod tests that demonstrate that vestibular defects are not observed in heterozygous KI mice by 7 months of age.

The manuscript is greatly improved from its original version and is worthy of publication in PLOS Genetics

Reviewer #2: This revised manuscript has addressed most of my previous concerns and is much improved. My only remaining concern is the interpretation of some of the results relating to the Pou4f3 -/+ mice as compared to the �/+ mice and the strong conclusions that the similarity between the mice is conclusive evidence that dominant mutations in Pou4f3 cause hearing loss as a result of haploinsufficiency.

The statement on line 187 is misleading, “The heterozygous knockout Pou4f3(-/+) mice displayed similar auditory phenotypes as the Pou4f3(Δ/+) mice (Fig 3)”. The auditory phenotype of the Pou4f3 (-/+) mice is not similar to the Pou4f3(�/+) mice. It appears to be less severe. This point is important as the manuscript is making a claim that Pou4f3 mutations cause hearing loss as a result of haploinsufficiency, which is in conflict with other mouse models and findings in the field. Thus, it is critical that the authors report on the results very precisely without over-stating the conclusions and interpretations.

One thing to consider is that possibility that both haploinsufficiency and a dominant negative activity are at play. The relatively protracted phenotype of the -/+ mice may indicate that haploinsufficiency is the cause of age-related high-frequency hearing loss in the mice with a complete loss of one copy of the gene whereas a dominant negative phenotype is the cause of the more severe hearing loss in the �/+ mice, where generation of a truncated protein may interfere with the function of the wild-type protein, for example.

Reviewer #3: As an attachment

**Have all data underlying the figures and results presented in the manuscript been provided?**

Reviewer #1: Yes

Reviewer #2: Yes

Reviewer #3: Yes

PLOS authors have the option to publish the peer review history of their article (what does this mean?). If published, this will include your full peer review and any attached files.

Reviewer #1: No

Reviewer #2: No

Reviewer #3: No

---

## [Editor Report · Decision Letter 2]

10 Aug 2020

Dear Dr Wan,

We are pleased to inform you that your manuscript entitled "Aldh inhibitor restores auditory function in a mouse model of human deafness" has been editorially accepted for publication in PLOS Genetics. Congratulations!

Yours sincerely,

Lisa Cunningham, Ph.D.

Guest Editor

PLOS Genetics

Gregory Barsh

Editor-in-Chief

PLOS Genetics

Comments from the reviewers (if applicable):

**Data Deposition**

http://datadryad.org/submit?journalID=pgenetics&manu=PGENETICS-D-20-00568R2

**Press Queries**

---

## [Editor Report · Acceptance letter]

18 Sep 2020

PGENETICS-D-20-00568R2

Aldh inhibitor restores auditory function in a mouse model of human deafness

Dear Dr Wan,

We are pleased to inform you that your manuscript entitled "Aldh inhibitor restores auditory function in a mouse model of human deafness" has been formally accepted for publication in PLOS Genetics! Your manuscript is now with our production department and you will be notified of the publication date in due course.

With kind regards,

Jason Norris

PLOS Genetics

On behalf of:
